# Characterization and Optimization of PLA Stereocomplexed Hydrogels for Local Gene Delivery Systems

**DOI:** 10.3390/polym11050796

**Published:** 2019-05-03

**Authors:** Kwei-Yu Liu, Daniel G. Abebe, Elizabeth Rachel Wiley, Tomoko Fujiwara

**Affiliations:** Department of Chemistry, The University of Memphis, 213 Smith Chemistry Bldg, Memphis, TN 38152, USA; kliu2@memphis.edu (K.-Y.L.); dgabebe@gmail.com (D.G.A.); erwiley@memphis.edu (E.R.W.)

**Keywords:** hydrogels, micelles, gene delivery, polylactide, stereocomplex, polyethylenimine

## Abstract

Localized gene delivery still remains as a challenging therapeutic method due to the multiple hurdles to overcome. One of the significant factors is a development of a matrix to carry and safely deliver genes at the local site in a controlled manner and then exit and disintegrate harmlessly. This report describes the structural and mechanistic studies on the in-situ forming hydrogels composed of the PEI/DNA multi-layered micelles to apply for gene therapy. The stereocomplexation-driven hydrogel systems from the DNA-loaded and DNA-free PLA-PEG-PLA triblock copolymer micelles that include enantiomeric polylactide blocks exhibited a sol-to-gel transitions between room and body temperatures. These hydrogels have well-described structure and compositions, and improved mechanical properties. Furthermore, the investigation of their degradation profiles and chemical analysis indicated the faster acidic degradation and stepwise degradation process of these micelle–hydrogel systems.

## 1. Introduction

Localized gene therapy approaches have found increasing popularity in the fields of tissue engineering, immune therapy, localized disease therapy, long-term depots for drugs, and localized site production “factories” for proteins [1,2,3]. For localized therapies, hydrogels have been utilized as coating materials for implants, intravascular stents, thin films (membrane), porous scaffolds, and in-situ forming hydrogel depots. Traditionally, hydrogels are produced outside of the body (after incorporating genetic material), and then implanted into the body. The disadvantage of such approach is that the matrix must be implanted through surgical means. Recently, the development of injectable hydrogels, which undergoes solution to gel transformation in the body has attracted considerable attention. Some of the advantages of these injectable delivery systems include minimal invasiveness and surgery related complication, and the materials can be molded to fit specific shapes and crevices [4,5]. In particular, temperature responsive hydrogels are highly desirable systems because the well-regulated physiological temperature is the only stimulus needed for sol-to-gel transition. 

The encapsulation and delivery of pDNA and siRNA have been reported using physically crosslinkable injectable hydrogels derived from natural polymers such as alginate, fibrin and gelatin [6,7,8,9,10,11,12]. The slow degradation properties and uncontrolled release profiles for these hydrogels have promoted researchers to develop block copolymer based injectable hydrogels. Some of the commonly utilized physically crosslinkable copolymers include the bioresorbable hydrophilic and hydrophobic blocks such as polyethylene glycol (PEG), polylactides (PLA), poly(lactide-*co*-glycolide) (PLGA), and polypropyrene glycol (PPG); for example, PLA-PEG-PLA, PLGA-PEG-PLGA, and Pluronic^®^ [13,14,15]. To improve their low mechanical strength typically associated with physical crosslinking, the chemically crosslinkable systems have been often used. Reactive functional groups such as fumarate, methacrylate, and acrylates are typically conjugated to gelling block copolymer or hydrophilic homopolymer systems (PEG is a commonly utilized hydrophilic homopolymer) [16,17,18,19,20]. Although chemical functionalization yields robust hydrogels, there are serious safety and biocompatibility concerns. The reactive cross-linkable groups and their degradation byproducts are known to cause localized toxicity and necrosis [21]. For the hydrogel systems incorporate the genetic material (pDNA, siRNA, etc.) in their naked form, the mechanism for loading of the genetic material relies simply on physical entrapment during the sol-to-gel transition process. These systems typically show low therapeutic efficiency due to lack of ability to retain the gene inside and prevent rapid and uncontrolled diffusion. Moreover, the released genetic material has low in-vivo half-life due to degradation by nucleases and inability to successfully cross the cell membrane [22]. To overcome these drawbacks, researchers have used cationic blocks such as polyethyleneimine (PEI) and poly(2-dimethylaminoethyl methacrylate) (PDMAEMA) to condense DNA into a polyplex particle and entrap it within the hydrogel matrix [23,24,25,26,27]. The main hurdle in the development of injectable hydrogels for the localized gene therapy is the fine balance needed between high mechanical strength and biocompatibility. Crystallization of polymer chains is an attractive alternative to a relatively toxic chemical crosslinking in the production of an injectable hydrogel with biologically relevant mechanical strength [28,29]. In particular, the stereocomplex crystal is produced through physical association of enantiomeric blocks of a thermo-gelling system (crosslinking junction). This crystal has relatively high thermal/physical stability, which allows it to act as a semi-permanent crosslinker similar to that produced by chemical methods [30].

In our previous work, the combination of triblock copolymers, PLLA-PEI-PLLA and PLLA-PEG-PLLA, has shown to produce nano-sized three-layered micelles (3LM), where the DNA/PEI polyplex is localized within a hydrophobic capsule [31]. The 3LM with relatively less toxic low MW linear-PEI has incorporated high concentrations of DNA. Additionally, 3LM proved to possess high stability in neutral pH, while its encapsulated payload was released through a pH mediated trigger allowing for controlled release properties. We further reported the initial studies of the thermo-gelling system between the DNA-loaded 3LM and the stereoisomeric block copolymer, PDLA-PEG-PDLA [32]. Overall, these systems have shown a great potential as non-viral gene delivery vectors. In this paper, we focus on the relationships of the 3LM compositions and their hydrogel properties to understand the role of PLA stereocomplexation in the system. The sol-to-gel transition behavior as a function of block MW and polymer composition in 3LM is studied. Additionally, to precisely tune the PLA stereocomplex derived hydrogels, the chemical and physical mechanisms of gel forming and degradation processes are investigated.

## 2. Materials and Methods

### 2.1. Materials

The monomers, l-lactide and d-lactide with the trademarked names of PURASORB L and PURASORB D, respectively, were purchased from Purac Biochem (Gorinchem, The Netherlands). PEG with a number-average molecular weight (*M*_n_) of 2000 and 3350 Da, the monomer 2-methyl-2-oxazoline (2-MeOx) 98%, *trans*-1,4-dibromo-2-butene (99%), tin (II) 2-ethylhexanoate (SnOct_2_), di-*tert*-butyl dicarbonate (ReagentPlus, 99%), trifluoroacetic acid (TFA), Salmon sperm DNA, chlorobenzene (extra dry), methanol (extra dry), acetonitrile (anhydrous, >99.8%), and toluene (extra dry) were purchased from Sigma Aldrich (St. Louis, MO, USA). Regenerated cellulose dialysis tube with MWCO (3.5–5k) was purchased from Spectrum Labs (Rancho Dominquez, CA, USA). The purification and storage procedures for all monomers, catalyst, macroinitiators and solvents used in this study can be found in our previous reports [29,31]. 

### 2.2. Measurements

Nuclear magnetic resonance (NMR) spectroscopy was used to determine the chemical structures and calculate the number-average molecular weight (*M*_n_) for all block copolymers. ^1^H NMR was performed on a Varian 500 MHz instrument at room temperature with CDCl_3_ and DMSO-d_6_ as solvents. Solid-state ^13^C NMR experiment of lyophilized hydrogel samples was performed through CP/MAS method on JEOL-ECZ 400 MHz spectrometer with a ^13^C operating frequency of 100 MHz and a spin rate of 10 kHz. The number- and weight-average molecular weights (*M*_n_, *M*_w_) and polydispersity of block copolymers was determined by gel permeation chromatography (GPC) on a Shimadzu LC-20AD with two Jordi DVB 500 Å (250 mm × 10 mm) columns calibrated with polystyrene standards at 35 °C, and THF was used as the mobile phase with a flow rate of 1.0 mL/min. The hydrodynamic diameter of micelles was determined by a dynamic light scattering (DLS) technique on a Zetasizer Nano ZS90 (Malvern Panalytical Ltd., Westborough MA, USA) at concentrations of 0.01, 0.1 and 1% at room temperature in triplicate. Mechanical properties of the resulting hydrogels were characterized on a TA AR 550 rheometer (TA Instrument, New Castle, DE, USA) using a cone-and-plate geometry with a 4° cone angle, 40 mm diameter plate, and 61 mm Gap. Rheological measurements were conducted as a frequency sweep from 0.01 to 40 Hz at 37 °C. Crystal structure was analyzed by wide angle X-ray scattering (WAXS) on a Bruker AXS D8 advance X-ray diffractometer for flash frozen/lyophilized powder samples. 

### 2.3. Synthesis of Triblock Copolymers

The facile and controlled synthesis of the ionic triblock copolymers PLLA-PEI-PLLA (MW: 1700-2000-1700) using an amine protection/de-protection approach can be found in our previous report [31]. Briefly, the PEI precursor α,ω-dihydroxy poly(2-methyl-2-oxazoline) (PMOx) was prepared by the ring opening polymerization (ROP) of 2-methyl-2-oxazoline (2-MeOx) using the bifunctional initiator *trans*-1,4-dibromo-2-butene. Linear PEI was obtained through base hydrolysis of the amide functional groups of PMOx. The secondary amines of PEI were then fully reacted with excess di-tert-butyl dicarbonate to obtain *N*-Boc protected PEI (PEI-*N*-Boc). The hydroxyl end-capped homobifunctional PEI-*N*-Boc, was used as a macroinitiator for the living ROP of L-lactide using the organo-metallic catalyst Sn(Oct)_2_ to yield the triblock copolymer PLLA-(PEI-*N*-Boc)-PLLA. The *N*-Boc protecting groups were then deprotected using TFA to yield the desired ionic triblock copolymer PLLA-PEI-PLLA. The amphiphilic triblock copolymers, PLLA-PEG-PLLA and PDLA-PEG-PDLA were prepared by the ROP method, using the macroinitiator PEG-diol and the monomers L-lactide and D-lactide, respectively. The complete synthesis procedure for the preparation of the amphiphilic triblock copolymers can be found in our previous report [29].

### 2.4. DNA Loaded 3LM Micelle Preparation

The detailed method for the formation of unique 3-layered micelle (3LM) system with the ability to entrap high MW DNA using a dual encapsulation approach was reported previously [31]. Briefly, the preparation of the DNA loaded 3LM is divided into two consecutive steps. Step 1, encapsulation via organo-micelle: the ionic triblock copolymer PLLA-PEI-PLLA (inner polymer) was dissolved in 1 mL DMSO and equilibrated at room temperature. Bulk DNA (1 mg) in 200 µL of pure water was gradually added to the polymer/DMSO solution under vigorous stirring. The solution was allowed to equilibrate at room temperature for 1 h under gentle stirring and was diluted with 4 mL of THF. The solution was dialyzed against THF using dialysis tubes (MWCO of 3.5–5 kD) to obtain the DNA loaded organo-micelles. Step 2, aqueous stabilization via an amphiphilic layer: To the organo-micelle/THF solution, the amphiphilic triblock copolymer, PLLA-PEG-PLLA (outer polymer, MW: 800-2000-800), was added to obtain a homogenous transparent solution. The mixed solution was then added dropwise to a 10 mL pure H_2_O under vigorous stirring. The solution was placed under a gentle stream of compressed air to firstly evaporate the organic solvent THF, followed by concentrating the aqueous solution to the desired volume to obtain the targeted concentration of the 3LM solution. To study and optimize the properties, 3LMs with different *inner:outer* polymer weight ratios were prepared.

PDLA-PEG-PDLA flower type micelle (D-micelle) was also prepared to use for stereocomplexed hydrogel formation. PDLA-PEG-PDLA (MW: 800-2000-800) in THF (20–40 *w*/*v*%) was added dropwise in DI water under vigorous stirring. THF and excess water were evaporated to obtain predetermined concentration of D-micelle solutions. Alternatively, the asymmetric PDLA-PEG-PDLA micelles with mixture of long/short PEG blocks (D-micelle-(%)S) were prepared [29]. The long-PEG copolymer, PDLA-PEG-PDLA (MW: 800-3350-800) and the short-PEG copolymer, PDLA-PEG-PDLA (MW: 800-2000-800) were weighed with the ratio of 100:0, 70:30, 50:50, 30:70, and 0:100 (%). Each mixture was fabricated into micelle form by same method above.

### 2.5. Stereocomplexed Hydrogel Preparation

The hydrogel formation procedure is similar to that reported for the standard PLA-PEG-PLA associative micelle network [29] and 3LM-hydrogel network [32,33] utilizing the PLA stereocomplexation. Controlled hydrogel formation was achieved by blending 3LM and counter-gelling PDLA-PEG-PDLA micelle solutions. The separately-prepared micelle solutions with matched concentration (for example, both solutions can be 20 wt % to obtain 20 wt % hydrogel) were mixed at adjusted volumes which were calculated in order to contain 1:1 ratio of PLLA-PEG-PLLA in 3LM and PDLA-PEG-PDLA in D-micelle with ultrasonic wave applied at 4 °C in a scintillation vial. The mixture was allowed to sonicate at 4 °C for 30 min or until a homogeneous solution is obtained. The vial is then transferred to a temperature-controlled circulating water bath and the temperature of the bath is gradually increased with 1 h hold at each temperature interval. The vial tilting method was used to determine sol to gel transition behavior, if the mixture flowed then it was reported as a solution and if it did not flow for at least 8 s it was reported as a gel [24,34].

### 2.6. Hydrogel Degradation

In vitro degradation study was conducted to simulate the degradation in physiological environment. The *inner:outer* polymer ratio within 3LM was set to 1:10, and the concentrations of 3LM and D-micelle-S were set to 20 wt %, as optimal setting based on the result from earlier experiments. After the hydrogel was formed, 200 mg each of the hydrogel was placed into 16 vials. Seven of those vials contained the hydrogel, 1 mL of 100 mM sodium acetate buffer with pH 4.5, and 1 mg of proteinase K. The next seven vials contained hydrogel, 1 mL of 10 mM Tris HCl with pH 7.4, and 1 mg of proteinase K. The fourteen vials were then placed in an incubator at 37 °C. The last two vials of hydrogel were dried with nitrogen gas, weighed, and then analyzed to determine the water content in the original hydrogel. Each vial for both pH buffer solutions was removed from the incubator at 3 h, 10 h, 1 day, 3 days, 7 days, and 14 days for the degradation analysis. The supernatant in each vial was decanted into a different vial, lyophilized, weighed, and then analyzed. Three different sets of this degradation test were performed.

## 3. Results and Discussion

### 3.1. Synthesis of Block Copolymers

The triblock copolymers PLLA-PEI-PLLA, PLLA-PEG-PLLA, PDLA-PEG-PDLA and P(DL)LA-PEG-P(DL)LA were prepared as previously reported [31]. Although the controllable ring-opening polymerization (ROP) of cyclic lactide monomers readily produced well defined triblock copolymers, only synthesis of PLLA-PEI-PLLA required a multi-step route to yield the comparative quality of copolymer. In general, the synthesis of PEI/PLA block copolymers has considerable difficulties such as incompatible solubility of PEI with that of lactide or PLA, aggressive nucleophilic nature of PEI amines, and low reactivity of PLA/PEI macromolecular coupling reactions. By utilizing the amine-protection chemistry, we were able to develop a facile and well-controllable synthetic procedure to yield well-defined block copolymers (Scheme 1). The PEI-*N*-Boc intermediate is soluble in a range of organic solvents and the potential initiation sights of the secondary amines are protected leaving only the terminal hydroxyl groups for ROP to occur. The final triblock copolymer of PLA-PEI-PLA was obtained by the mild deprotection of Boc groups using 50 wt % TFA/CHCl3 solution for 5 min. The deprotection step leads to protonation of the PEI block rendering a positive charge which is necessary for the complexation step with negatively charged DNA.

### 3.2. 3LM and Hydrogel Formation

Formation of DNA-loaded 3LM was previously confirmed by TEM in our report, where the stained DNA was observed in the core and surrounded by polymeric shells with micelle diameters 100–200 nm [31]. The compositions of 3LM and proposed structure of its stereocomplexed hydrogel are illustrated in Figure 1. The chain-exchange mechanism between enantiomeric PLLA-PEG-PLLA and PDLA-PEG-PDLA micelle solutions were previously reported [29]. To further study the mechanisms and optimize the properties of 3LM-hydrogels, 3LM with different *inner:outer* polymer ratios were prepared as shown in Table 1. Additionally, Table 1 lists the different PEG block length of PDLA-PEG-PDLA used for D-micelles. 3LM and D-micelles were prepared with various concentrations to examine the gel formation process.

### 3.3. Effect of Formulation on Micelle Size and Composition

The effect of the *inner:outer* polymer ratios on the 3LM particle size and structure was determined by DLS measurements. The hydrodynamic radius is given for the 3LM solutions from four *inner:outer* ratios investigated, 1:1, 1:5, 1:10 and 1:20 (Figure 2a). The “*inner*” value corresponds to the weight fraction of PLLA-PEI-PLLA, whereas the “*outer*” value corresponds to that of PLLA-PEG-PLLA which makes up the corona of 3LM. As agreed with the previous reported 3LM data, the 1:1 ratio formulation gave the particles with monodispersed sizes of 100–200 nm. However, with increasing *outer* ratio values in the 3LM formation process, a second peak at <50 nm was observed. The intensity of the second peak increased with increasing *outer* ratio, while the intensity of the original peak decreased. The second peak at sizes <50 nm corresponds to the simple flower-micelles composed of simple PLLA-PEG-PLLA triblock copolymers (L-micelle); and the hydrodynamic sizes observed are comparable to the micelles from triblock copolymers with same molecular weight [35]. Therefore, with increasing *inner:outer* ratios, the micellar solution is populated with two sets of micelle particles, 3LM (containing DNA) and the flower micelles (DNA free). The DLS data indicates the size of 3LM is not significantly affected by the increasing concentration of the PLLA-PEG-PLLA outer layer. Even though the excess of PLLA-PEG-PLLA outer polymers has a negligible effect on the 3LM self-assembly properties, the presence of L-micelles may affect the sol-to-gel transition behavior and gel properties considerably and that will be discussed below. We have also examined the 3LM size using the longer PLLA block of the outer polymer, PLLA-PEG-PLLA (1200-2000-1200) (Figure 2b). Interestingly, the thickness of hydrophobic intermediate layer created by both inner and outer PLLA blocks did not affect to the overall diameter of 3LMs.

### 3.4. Gelation Mechanism and Sol-to-gel Phase Transition

Thermo-responsive hydrogels were prepared using the stereocomplexation mechanism of enantiomeric polylactides. Stereocomplexed (physically crosslinked) hydrogels are attractive because they are free from potentially toxic cross-linkers typically required for chemically crosslinked hydrogels. Although the mechanical strength of physically crosslinked hydrogels are generally lower than chemically crosslinked hydrogels, the enantiomeric mixture of PLA-PEG-PLA micelles has shown significantly increased storage modulus of stereocomplexed hydrogels by controlling the polymer sizes and micelle packing parameters [28,29]. The hydrogel formation is driven by the stereocomplexation of the PLLA and PDLA blocks of the micellar solution, which is the result of the chain exchange between neighboring micelles. The inter-micelle chain exchange predominates at higher temperatures and concentrations, leading to an irreversible hydrogel. 

The insight of hydrogel formation from 3LMs was studied in this work. As depicted in Figure 1 above, the PDLA block is supplied by the blank D-micelles as one of the sources of stereocomplexation in the hydrogel formulations. Whereas, there are several sources the PLLA block can be obtained in the system. The intermediate hydrophobic layer of 3LM is composed of two sets of PLLA, one derived from the DNA complexed PLLA-PEI-PLLA copolymer and the second from the corona forming PLLA-PEG-PLLA copolymer. The stereocomplexation ability of each PLLA segment was investigated by replacing the corona forming copolymer (outer polymer) with the racemic and amorphous PDLLA-PEG-PDLLA which has no ability to form stereocomplex crystals during the 3LM-hydrogel preparation step. The presence/absence of the stereocomplexation diffraction peaks was determined through WAXS after lyophilized the mixtures of 3LM and D-micelle solutions. The 3LM containing PDLLA-PEG-PDLLA outer layer did not form a hydrogel after mixing with D-micelle solution at all temperatures and concentrations investigated. Justifying this result, we observed no stereocomplexation diffraction peak in the WAXS spectra for this mixture (Figure 3a) compared to the typical stereocomplex peaks observed in simple micelle mixture of PLLA-PEG-PLLA and PDLA-PEG-PDLA (Figure 3c). In contrast, the stereocomplex diffraction peaks at 2θ = 12° and 21° were clearly observed in the WAXS spectra of the hydrogel forming 3LM including semicrystalline PLLA-PEG-PLLA outer layer mixed with D-micelles (Figure 3b). Therefore, the PLLA blocks of the outer layer polymer are responsible for stereocomplexation event leading to hydrogel formation. For 3LM with increasing *inner:outer* ratios, 1:5, 1:10, 1:15 and 1:20, the stereocomplexation event is likely a combination of chain exchange between both 3LM and excess of blank PLLA-PEG-PLLA micelles and blank PDLA-PEG-PDLA micelles, leading to hydrogel formation.

The phase transition (sol-to-gel) diagrams for the different formulations are shown in Figure 4. The phase diagram of 3LM with the 1:1 *inner:outer* ratio mixed with D-micelle (Figure 4a) shows a linear decrease in the sol-gel transition temperature with increasing polymer concentration. Hydrogels with clinically relevant sol-gel transition temperatures, 37 and 25 °C can be obtained at 20 and 25 wt %, respectively. The phase diagram of the different *inner:outer* formulation in 3LM is shown in Figure 4b. The concentration of all the solutions was kept constant at 20 wt % and thus only the effect of the increasing population of excess PLLA-PEG-PLLA micelles was investigated. The phase diagram shows a linear decrease in sol-gel temperature with increasing *inner:outer* ratio, indicating the incorporation of blank L-micelle contributes with the overall network formation (crosslinking). The 3LM-1/20 solution has the highest L-micelle population, and the sol-to-gel transition occurred at the lowest temperature that is comparable to the 3LM-blank mixture (50/50% L- and D-micelles). The chain exchange rate of PLA blocks between all micelles depends on the polymer and micelle compositions. The faster exchanges between simple L- and D-micelles for the systems with larger *inner:outer* ratios in 3LM would make the network structure even at the low temperature. Furthermore, the relatively large size of 3LM may not be suitable for the close micelle packing needed for robust hydrogel formation. The blank PLLA-PEG-PLLA micelles could be able to bridge the gap and allow for more crosslinking points. As a result, the hydrogels are readily formed at lower temperatures compared to the system without blank PLLA-PEG-PLLA micelle (i.e., 1:1 *inner:outer*). Overall, sol-to-gel transition temperature is precisely tunable with micelle concentration and composition (Figure 5).

### 3.5. Mechanical Strength

The mechanical strength of hydrogels is an important physicochemical property, which plays a crucial role in the utilization of the hydrogel for biomedical applications. The mechanical strength of the DNA loaded 3LM was characterized using rheological profiles. Figure 6 shows the storage (*G*′) and loss (*G*″) moduli of the 3LM hydrogels at *inner:outer* ratios of 1:1 and 1:10, and the control hydrogel from the simple PLLA-PEG-PLLA micelle; plotted immediately after mixing with the PDLA-PEG-PDLA micelle solution. All measurements were performed at 37 °C with a constant solution concentration of 20 wt %. The L+D blank micelles show the highest storage modulus, indicating that the presence of 3LM in the hydrogel network leads to a decrease in mechanical strength. The hydrodynamic sizes of the co-gelling 3LM and PDLA-PEG-PDLA micelles are considerably different. Thus, the inter-micelle distance, micelle packing density, micelle aggregation behaviors and chain exchange kinetics would be different than the L+D blank micelles. In our previous study, we utilized time-resolved small angle neutron scattering (TR-SANS) to investigate the effects of chain exchange kinetics on the formation of L+D stereocomplexed hydrogels [35]. Model dependent analysis of the scattering data suggested, (1) micelle volume increased due to increase in aggregation number following non-equilibrium chain exchange, (2) chain exchange rate was influenced by the size of the PEG corona, with larger PEG corona showing a delayed response in aggregation number increase and (3) micelles initially formed small and tight clusters, which then grew to larger clusters and eventually formed the hydrogel network. The storage modulus is considerably lower for the *inner:outer* ratio 1:1 formulation than the control in Figure 6. When the simple PLLA-PEG-PLLA micelles are incorporated into the gelling solution by increasing the *inner:outer* ratio, a clear increase in *G*′ is observed. The improvement in mechanical strength of the 1:10 *inner:outer* formulation can be attributed to increase in micelle packing density. The simple PLLA-PEG-PLLA micelles increase the crosslinking points, effectively bridging the gap that existed between 3LM and PDLA-PEG-PDLA micellar populations in the 1:1 *inner:outer* formulation.

### 3.6. Effect of PEG Block Length on the Hydrogels

We have previously studied the effect of PEG block length in PLA-PEG-PLA to the kinetics of gelation process and hydrogel properties [29]. Compared to the control hydrogel prepared from PLLA-PEG-PLLA (MW: 800-2000-800) and PDLA-PEG-PDLA (MW: 800-2000-800) simple micelle solutions, mixing the different copolymers with longer PEG block yielded different sol-to-gel transition temperatures and mechanical strength. The inter-micelle exchange of PLA chains for the micelles with short PEG polymer were found to be more rapid than those with long PEG polymer. The micelles made of the mixed PEG size copolymers (as illustrated in Figure 7) exchange PLA blocks with different rate and the sol-to-gel transition temperature changed linearly by the mixing ratios [29]. Moreover, the hydrogels from the short/long mixed PEG micelles exhibited remarkably higher modulus [29] 

In this study we mixed long-PEG copolymer, PDLA-PEG-PDLA (MW: 800-3350-800) to prepare D-micelles (Figure 7) and used for 3LM-hydrogel formation. Table 2 summarizes the observation of sol-to-gel transition temperatures and flow behavior of 3LM-hydrogels by tilt method. In agreement with our previous work, the PEG length affected to the gel formation and properties of 3LM-hydrogel system. The transition temperature is the highest when using D-micelle–30S (short-/long-PEG = 30:70 wt %), that also indicates the slower chain exchange of longer polymers. However, those gels with long/short mixed length micelles (50S and 30S) do not flow over 60 min and are apparently more robust than the hydrogel with D-micelle from PDLA-PEG-PDLA (800-2000-800) only.

### 3.7. Hydrogel Degradation Study

The progress of hydrogel degradation was monitored periodically by the weight loss of hydrogels in both neutral and acidic environment with addition of the enzyme. Proteinase K is known to selectively degrade PLLA with relatively low crystallinity. As shown in Figure 8, the hydrogel in the Tri-HCl buffer at pH 7.4 maintained a fairly consistent weight up to 2 weeks with 83% in average of the original weight. In contrast, the hydrogel in the acetate buffer at pH 4.5 showed a steady decline and 48% in average of the initial weight remained as a gel after 2 weeks. It should be noted that the hydrogels physically disintegrated after 2 weeks, and no reliable trend on the degradation profile was obtained. Separately, enzyme-free degradation samples exhibited relatively slower rate.

To study the degradation process of 3LM-hydrogel, the supernatants of the hydrogel test solutions after separation and lyophilization were analyzed by GPC and NMR. As the GPC curves of the accurately re-dissolved samples in Figure 9 show, the PEG (MW = 2000, polydispersity = 1.08) peak was detected after 24 h incubation in both neutral and acidic conditions with the presence of the enzyme. Therefore, it was speculated that some degree of PEG blocks was initially cleaved and released into the solution within one day regardless of the pH by enzymatic degradation at the junction with the PLLA blocks. However, the released components at neutral condition remained unchanged whereas the degradation in an acidic solution progressed in time (Figure 9b); the PEG peak appeared immediately, followed by the smaller molecular weight oligomers from PLLA block were released. Additionally, the higher molecular weight polymers were released into the buffer after 1 week. Those polymers, identified as PLLA-PEG-PLLA and PDLA-PEG-PDLA, are believed to be diffused into the solution as a result of the disintegration of the micelle–hydrogel matrix. Based on these results, the degradation process and mechanism of the 3LM-hydrogels are proposed as follows: (1) the enzyme, proteinase K hydrolyzed the PLLA chain at the PLLA-PEG junction (the outer most PLLA of the micelle core) and released some PEG blocks into the solution, (2) further enzymatic degradation of PLLA proceeded slowly within 3LM, and (3) physical disintegration of hydrogel resulted in the release of remaining block copolymers. Furthermore, acidic hydrolysis of the PLLA chain in pH 4.5 played a significant role to promote the degradation process.

The supernatants of hydrogel degradation samples were also analyzed by ^1^H NMR. The NMR spectra in Figure 10 show a PEG peak at 3.47 ppm and multiple peaks corresponding to CH_3_ and CH for PLLA and its degradation products such as lactic acid and oligomers in the buffer solutions at different pH after 1 and 7 days. The intensity of all spectra was normalized using internal reference peak. It revealed that the concentration of degraded PLLA products was significantly higher in the acidic solution (red) than in the neutral solution (green) after 1 week. For the supernatants after 1 day (blue) and 7 days (red) in the acidic solution, the PEG peak was detected since day 1, and the PEG peak (see the enlarged spectra) and degraded PLLA species increased significantly after 1 week. These data correlate to the GPC results and support the proposed degradation mechanism. In the neutral pH solution, there was no noticeable difference of degradation products of PLLA, such as lactic acid and oligomers, between 1 day (data not shown) and 1 week (green). Based on those results, it is hypothesized that the DNA can be released from the hydrogel faster in the acidic solution, and degradation of PLLA block, i.e., intermediate layer of 3LM. 

### 3.8. Hydrogel Analysis by Solid-State NMR 

To further investigate the PLA crystal structures in the hydrogel from L-/D-micelles, CP/MAS ^13^C NMR spectra were obtained for the polymer and lyophilized hydrogel samples at different stages. Figure 11 shows the spectra of PLLA-PEG-PLLA (800-2000-800) and stereocomplexed hydrogels (mixture of L- and D-micelles) which was immediately lyophilized, slowly air-dried, and treated in buffer at pH 7 and pH 4.5 for 5 day. The advantage of CP/MAS NMR is to identify the different polymer packing state. The spectrum of PLLA-PEG-PLLA (a) shows PLLA-methyl peak around 20 ppm, PLLA-methine peak around 72 ppm, PEG peak at 74 ppm, and PLLA-carbonyl peak around 175 ppm. Most of PLLA carbons display as multiple peaks due to the semicrystalline nature. For example, the inserted spectra are the expansion of the carbonyl peak region in which amorphous and crystal carbon peaks are distinctly separated [30]. The freshly made and lyophilized hydrogel from enantiomeric micelles of PLLA-PEG-PLLA (800-2000-800) and PDLA-PEG-PDLA (800-2000-800) (b) showed all three forms for the carbonyl carbon; homocrystal, amorphous, and stereocomplex crystal at 173, 174.5, and 176 ppm, respectively. This indicates that relatively low degree of chain exchange and stereocomplex formation still leads transformation of the micelle solution into a gel form. When the hydrogel was slowly dried up in air (c), we observed only distinguished peak of stereocomplexed PLA for carbonyl carbon. The hydrogels remained in buffer solutions after 5 days maintained good structure of stereocomplex PLA (d, e) although PEG and small PLA oligomers already started to release into the buffer solutions. Since both hydrolysis and biodegradation of PLA start from amorphous region, it is reasonable that stereocomplexed PLA remains longer in the hydrogel. Note that the small peaks at 30 and 184 ppm are attributed to the acetate buffer used for degradation test.

### 3.9. Gene Delivery Prospectives of 3LM-hydrogels

As mentioned in the introduction, 3LM as a DNA delivery carrier has been evaluated in the previous paper [31], and its stereocomplexed hydrogel has also shown the potentials in our preliminary study (summarized in Table 3) [32]. However, physical properties, structural transformation, and polymer compositions of the stereocomplexed hydrogel would significantly affect biological test results. The relationship between the micelle structures and gelation mechanism, gel strength, and degradation process revealed in this detailed and systematic study demonstrated the reproducibility and potential use of 3LM-hydrogels for optimized local gene delivery systems. 

## 4. Conclusions

PLA stereocomplexed hydrogel systems incorporating DNA-loaded 3LM were obtained and analyzed. The sol-to-gel transitions of the 3LM-hydrogels were observed to vary based on several factors of formulation conditions. The hydrogel systems with a sol-to-gel transition at the desired temperature were obtained by optimizing the polymer concentration in the system and weight ratio of *inner:outer* polymers (PLLA-PEI-PLLA/PLLA-PEG-PLLA) during the 3LM formation. Furthermore, the mechanical strength of the hydrogels improved through the use of PDLA-PEG-PDLA micelles with mixed PEG block length. The structure, properties, and mechanistic aspect of the 3LM-hydrogel system were well correlated with stereocomplexation behavior of enantiomeric PLA blocks. The degradation study revealed that the 3LM-hydrogel degraded fairly quickly in an acidic environment and remained stable in a neutral environment. The PEG blocks started to be released through enzymatic degradation at the junction of the block copolymer regardless of the pH, and subsequent PLLA degradation proceeded in the acidic solution. These results show the 3LM-hydrogel is a well-structured and tunable injectable non-viral gene vector that is readily prepared through stereocomplexation between enantiomeric PLLA/PDLA thermo-gelling systems.

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
