# Peer review of "Characterization and Optimization of PLA Stereocomplexed Hydrogels for Local Gene Delivery Systems"

_polymers, 2019, doi:10.3390/polym11050796_

Round 1
Reviewer 1 Report
In this article, Liu et. al. presented some additional data to complete the characterization and optimization of their already published stereocomplexed hydrogels.
This reviewer finds this article suitable for publication in the present form, as it further strengthens the reproducibility and potential use of an already published gene delivery hydrogel system.
Minor language edits are required: for example; on page 2, line 60; it must be "biologically relevant" instead of biological relevant. Similar errors are there in the manuscript.. make sure to correct them before publication.
Author Response
Thank you very much for evaluating the importance of detailed studies on our hydrogel system for the reproducibility and potential use. That is what exactly we have aimed to conduct these studies.
We have re-checked the manuscript for any grammatical and language problems and revised thoroughly.
Reviewer 2 Report
In this manuscript, the authors have synthesized, optimized and characterized PLA stereocomplexed hydrogels for local gene delivery systems. This manuscript is reviewed in term of 1) the novelty of materials; 2) characterization of the materials; and 3) the biological performance. 1. The materials are of good novelty. It is a good idea to incorporate PEI segment into PLA block copolymers and form the biodegradable stereocomplexed hydrogels for gene delivery. 2. The authors have also fully characterized the chemical and physical properties of the polymers and the hydrogels. However, since this hydrogel is prepared for gene delivery, it is a pity that the authors haven’t performed any relevant experiments and provided any information. 3. And it is also a pity that the authors don’t provide any biological performance for these materials for biomedical applications. I suggest the authors to add some experiments (e.g. Gel retardation analysis to confirm the ability of the material in carry and maintain gene materials; cytotoxicity experiments to confirm the biocompatibility of the materials; any experiments indicating the successful gene delivery)Author Response
Thank you very much for pointing out the novelty and unique physical properties of our stereocomplexed hydrogel system. As we stated in Introduction, page2, line 65-72 (copied below), this DNA-loaded 3LM system has been well evaluated and published in our previous papers. We also have reported preliminary biological data on the hydrogel system. However, we thought this unique thermos-gel system must be studied in more details to understand its gelation mechanism and degradation process, and decided to report in this manuscript. We understand it was ideal to report this article first, and not mentioning biological outcomes is unusual as a complete paper. Thus, we added the discussion to summarize the gene delivery capability and materials evaluation, and related to the findings in this study in page 13.